# Virological Aspects of COVID-19 in Patients with Hematological Malignancies: Duration of Viral Shedding and Genetic Analysis

**DOI:** 10.3390/v17010046

**Published:** 2024-12-31

**Authors:** Asma Themlaoui, Massimo Ancora, Kais Ghedira, Yosra Mhalla, Manel Hamdoun, Maroua Bahri, Lamia Aissaoui, Raihane Ben Lakhal, Adriano Di Pasquale, Cesare Camma, Olfa Bahri

**Affiliations:** 1Laboratory of Microbiology and Biochemistry (LR16SP01), Aziza Othmana Hospital, University Tunis El Manar, Tunis 1068, Tunisia; 2National Reference Centre for Whole Genome Sequencing of Microbial Pathogens: Database and Bioin-Formatic Analysis (GENPAT), Istituto Zooprofilattico Sperimentale dell’Abruzzo e del Molise, 64100 Teramo, Italy; 3Laboratory of Bioinformatics, Biomathematics and Biostatistics (LR20IPT09), Pasteur Institute of Tunis, Tunis 1002, Tunisia; 4Hematology Department, Aziza Othmana Hospital, University Tunis El Manar, Tunis 1068, Tunisia

**Keywords:** hematological malignancies, SARS-CoV-2 infection, whole-genome sequencing, viral shedding

## Abstract

Coronavirus disease 2019 (COVID-19) has been associated with a significant fatality rate and persistent evolution in immunocompromised patients. In this prospective study, we aimed to determine the duration of excretion of severe acute respiratory syndrome coronavirus 2 (SARS-CoV-2) in 37 Tunisian patients with hematological malignancies (40.5% with lymphoma and 37.8% with leukemia). In order to investigate the accumulation of viral mutations, we carried out genetic investigation on longitudinal nasopharyngeal samples using RT-PCR and whole-genome sequencing. Patients’ samples were collected until the RT-PCR results became negative. SARS-CoV-2 infection was symptomatic in 48.6% of cases with fever, and cough was symptomatic in 61% of cases; the mortality rate was estimated to be 13.5%. The duration of viral RNA shedding ranged from 7 to 92 days after onset; it exceeded 18 days in 79.4% of cases. An intermittent PCR positivity was observed in two symptomatic patients. Persistent PCR positivity, defined as the presence of viral RNA for more than 30 days, was found in 51.4% of cases. No significant differences were observed for age, sex, type of hematological malignancy, or COVID-19 evolution between this group and a second one characterized by non-persistent PCR positivity. Lymphopenia was an independent predictor of prolonged SARS-CoV-2 RNA detection (*p* = 0.04). Three types of variants were detected; the most frequent was the Omicron. Globally, the mean intra-host variability in the SARS-CoV-2 genome was 1.31 × 10^−3^ mutations per site per year; it was 1.44 × 10^−3^ in the persistent group and 1.3 × 10^−3^ in the non-persistent group. Three types of mutations were detected; the most frequent were nucleotide substitutions in the spike (S) gene. No statistically significant difference was observed between the two groups as to the type and mean number of observed mutations in the whole genome and the S region (*p* = 0.650). Sequence analysis revealed the inclusion of one to eight amino acid-changing events in seventeen cases; it was characterized by genetic stability from the third to the twentieth day of evolution in six cases. For the two patients with intermittent PCR positivity, sequences obtained from samples before and after negative PCR were identical in the whole genome, confirming an intra-host evolution of the same viral strain. This study confirms the risk of persistent viral shedding in patients with hematological malignancies. However, persistence of PCR positivity seems to be correlated only with a continuous elimination of viral RNA debris. Additional studies based on cell culture analysis are needed to confirm these findings.

## 1. Introduction

The severe acute respiratory syndrome coronavirus 2 (SARS-CoV-2) is a positive-stranded RNA virus belonging to the *Coronaviridae* family; it is characterized by high genetic variation, with more than 25,000 mutations previously reported [1]. From the “wild-type” virus described in early 2020, at least 12 important variants emerged; five of them (Alpha, Beta, Gamma, Delta, and Omicron) were classified by the World Health Organization (WHO) as variants of concern [2]. The emerging variants have increased transmissibility, potentially higher pathogenicity, and reduced sensitivity to neutralization by therapeutic monoclonal antibodies [3]. Implicated mutations are mostly described in the S region and are linked specifically to variant [4]; most of them are mutations that have been associated to the Omicron variant: high infectivity (69–70 deletion), relapse to immunological response (E484K, N440K, G446S), high transmissibility (L452R, P681H/R, T478K, Q493R, and Q498R), or impact on host interactions [3,5,6,7,8].

SARS-CoV-2 can be excreted in different biological liquids, such as respiratory samples, urine, and stool [9,10]. The median duration of viral RNA shedding in respiratory samples is 18.4 days; however, it can be longer, up to 31 days from illness onset [9]. The persistence of viral excretion is not well defined yet; it was reported with some mutations in the S region, like T478K, Q498R, N501Y, D614G, H655Y, P681H, N764K, D796Y, D796Y, Q954H, and N969K [11]. However, the role of these mutations in viral persistency remains unclear. Persistent viral shedding was also commonly described with a severe illness or with high levels of immunodeficiency; it could potentially accelerate genomic viral evolution [12,13]. A six-month-period was reported for persistent PCR positivity after infection in a 61-year-old female patient diagnosed with stage IVB non-Hodgkin diffuse large B-cell lymphoma [12]. So, immunocompromised patients can possibly serve as reservoirs to maintain viral circulation and to select new mutants with different levels of infectivity and contagiousness [14]. It poses, therefore, the additional threat of extended transmissibility and selection pressure that could generate immune-evading mutations.

Since its emergence, Coronavirus Disease 2019 (COVID-19) has been associated with high mortality rates and severe evolution in patients with advanced age and/or comorbidities like diabetes, hypertension, or chronic lung disease [15]. It was particularly severe in patients with hematological malignancies; mortality rates reached 21% in patients with lymphoid malignancies, and can be over 54% in myeloma [14,16,17,18,19]. In Tunisia, up to now, no data are available about the infection severity, genetic variability, or duration of viral RNA shedding in the general population and, in particular, patients with hematological malignancies. Since detection of the first Tunisian case of COVID-19 on 3 March 2020, only epidemiological studies were performed, with subsequent descriptions of circulating variants and the assessments of the impacts of national prevention strategies used. These strategies have been based on international recommendations. For patients with hematological malignancies, SARS-CoV-2 screening was performed systematically by real-time polymerase chain reaction (RT-PCR) in nasopharyngeal swabs before hospitalization and before starting chemotherapy. No control by PCR was then performed for positive patients. All positive cases were isolated, at first, for 14 days; the duration of isolation was then lowered to 5 days according to international recommendations from the World Health Organization (WHO) [20].

We, therefore, realized this study with the aim of estimating the rate of SARS-CoV-2 persistence in infected Tunisian patients suffering from hematological malignancies and to characterize its virological aspects.

## 2. Materials and Methods

### 2.1. Studied Population

This prospective study was conducted from May 2021 until August 2022. It included all patients hospitalized for management of hematologic malignancy and for whom a SARS-CoV-2 infection was confirmed by RT-PCR performed in nasopharyngeal samples. Molecular detection was performed before hospitalization for SARS-CoV-2 infection screening or when COVID-19 was suspected in hospitalized patients. The medical background and clinical course during the COVID-19 illness were identified for these patients from electronic medical records. Monitoring of SARS-CoV-2 infection in these patients was performed by RT-PCR on longitudinal nasopharyngeal samples collected weekly for each patient until the results were negative; one sample per week had been planned for each patient.

### 2.2. SARS-CoV-2 RNA Tests

At first, viral RNA was extracted from 400 µL of sample using an automated nucleic acid extraction system (croBEE NA16 Nucleic Acid Extraction System Plus, GeneProof-UK, Dolni Herspice, Czech Republic). Amplification was performed using the commercial assay Allplex Sars-CoV-2 assay (Seegene, Seoul, Republic of Korea) as recommended by the manufacturer. The assay targeted RdRP, N, and E genes, and a sample was confirmed positive for SARS-CoV-2 if reaction growth curves crossed the threshold line within 38 cycles (Ct cutoff ≤ 38.0) for the E gene and both RdRp and ORF, or either RdRp or ORF. Positive and negative controls were used at each run. The absence of inhibitors in samples was verified by using an internal control (the RNase P gene).

### 2.3. Whole-Genome Sequencing Protocol

This part of the study was performed in collaboration with the National Reference Center for Whole-Genome Sequencing of Microbial Pathogens—database and bioinformatic analysis center based at the Istituto Zooprofilattico Sperimentale (Teramo, Italy). Next-generation sequencing was performed on all positive samples. Whole-genome sequencing (WGS) was carried out using the COVIDSeq Test (Illumina Inc., San Diego, CA, USA) following the manufacturer’s instructions. Briefly, reverse transcriptase and random hexamer primers were used to synthesize first-strand cDNA from 8.5 µL of extracted viral RNA. The SARS-CoV-2 genome was then amplified using two separate PCR reactions (COVIDSEQ Primer Pool 1 and COVIDSEQ Primer Pool 2). Library preparation involved tagmentation and adapter ligation using IDT (Integrated DNA Technologies) for Illumina (PCR Indexes Set A–D). Purification was performed with Ampure XP beads, and quantification was conducted using the Qubit 2.0 Fluorometer (Life Technologies, Carlsbad, CA, USA) with the Qubit dsDNA HS Assay Kit (Invitrogen, Carlsbad, CA, USA). The library size was then determined using the 4200 TapeStation (Agilent Technologies, Santa Clara, CA USA). Finally, the libraries were diluted to 2 nM, pooled, and loaded at a final concentration of 750 pM, with 2% PhiX internal library control (750 pM), onto the NextSeq 2000 platform (Illumina Inc., San Diego, CA, USA) using the NextSeq 2000 P1 Reagents Cartridge.

### 2.4. Bioinformatic Analysis

Raw sequence data generated by the NextSeq 2000 were uploaded to the GENPAT platform (https://github.com/genpat-it; accessed on 15 November 2022). For quality control and trimming, the fastQ files were filtered using FastQC version 0.11.9 and Trimmomatic version 0.39 to remove low-quality reads and adapters. After trimming, sequences were aligned to the original Wuhan-Hu-1 reference genome (Acc no. NC_045512) by the BWA tool for mapping and the consensus sequences were obtained using iVar. The online tools Nextclade v.2.14.1, obtained via the web: https://clades.nextstrain.org/ (accessed on 2 December 2022), and the Pangolin (version v4.2) PANGO tool, as obtained via the web: https://pangolin.cog-uk.io/ (accessed on 2 December 2022), were also run on those consensus sequences to confirm the occurrent SARS-CoV-2 lineage generated by the GENPAT platform [21]. The FASTA files obtained were uploaded to the online tool MAFFT (Multiple Alignment using Fast Fourier Transform) via the web: https://mafft.cbrc.jp/ (accessed on 15 December 2022). Multiple alignments were manually edited using BioEdit and BLAST via the web: https://blast.ncbi.nlm.nih.gov/. Mutations were reported using the online tool Coronavirus Antiviral and Resistance Database (A Stanford HIVDB Team website) via the web: https://covdb.stanford.edu/sierra/sars2/by-sequences/(accessed on 10 March 2023). For each patient, the successive samples were compared to the first sequence as a reference to study all the genetic modifications following the infection by SARS-CoV-2.

### 2.5. Clade and Lineage Assignment

Whole-genome sequences, in Fasta format, were used for clade and lineage assignment using the online tools Nextclade (https://clades.nextstrain.org/) and Pangolin (https://pangolin.cog-uk.io/) (accessed on 2 March 2023).

The Fasta files were used to build a phylogenetic tree using the online tool Nextclade v.2.14.1 via the web: https://clades.nextstrain.org/tree (accessed on 5 March 2023).

### 2.6. Statistical Analyses

Descriptive statistics were expressed as mean values for continuous variables and as numbers (percentages) for categorical variables. To estimate significant differences, Fisher’s exact and chi-squared tests for trend were used for categorical variables, while the Mann–Whitney and Kruskal–Wallis tests were used for continuous variables. Statistical analyses were performed with the SPSS software package for Windows (version 23.0, SPSS Inc., Chicago, IL, USA). Bonferroni’s correction (multiple comparison corrections) was applied in order to obtain statistically significant *p*-values (<0.05).

## 3. Results

### 3.1. Characteristics of Patients

During the study’s period, 37 patients were included; the mean age was 36 years. Fifteen patients (40.5%) had lymphoma and fourteen (37.8%) had leukemia, the majority of them with acute lymphoid leukemia (ALL). According to age, type, and stage of disease malignancy, three types of treatment were administered to patients: chemotherapy alone (eleven cases), an association of chemotherapy and corticoids (eighteen cases), or immunotherapy combined with chemotherapy and high doses of corticoids (five cases). Comorbidities were reported in 15 cases: hypertension in eight cases and diabetes in four cases; the other comorbidities detected were chronic lung diseases (three cases), chronic hepatitis B (two cases), and hypothyroidism (two cases). At least two comorbidities were associated in seven cases. The demographic and clinical features of patients are shown in Table 1.

SARS-CoV-2 infection was symptomatic in 48.6% of cases and characterized by fever and cough in 61% of cases. Severe evolution ending in death was observed in five cases, whose ages ranged from 56 to 67 years; the mortality rate was estimated to be 13.5%. Patient management was carried out according to the national guidelines established by expert committees and published by the National Agency for Health Evaluation and Accreditation (INEAS) in 2020 as “Guide Parcours du patient suspect ou atteint par le COVID-19” [22]. Briefly, the treatment was based on symptoms, without using monoclonals, immunoglobulins, or antiviral drugs. It depended on severity and involved oxygen therapy, antibiotics, and painkillers [22]. Vaccination status against SARS-CoV-2 was not recorded.

### 3.2. Persistence of SARS-CoV-2 Shedding

A total of one hundred and six positive nasopharyngeal samples were investigated; at least two samples were performed by the patient. Nineteen, nine, and six patients benefited from two, three, and four samples, respectively. For three patients, 5 (n = 2) or 10 (n = 1) samples were collected. The duration of viral RNA shedding ranged from 7 to 92 days after onset (Figure 1). It exceeded 18 days for 79.4% of patients (n = 27); the mean duration was 34 days. For three patients who died, the duration of viral shedding was not estimated. Two symptomatic patients were characterized by intermittent PCR positivity, which was observed through Day 14 and Day 39 (Figure 2).

In the absence of established criteria, we adopted the following definition for the persistence of SARS-CoV-2 shedding: “a presence of viral RNA for more than 30 days after illness onset for symptomatic patients and after the firs tpositive sample for asymptomatic cases”. Persistent PCR positivity was detected in 51.4% of cases (n = 19). Table 2 reports the demographic and clinical features of patients with persistent PCR positivity as well as a statistical comparison to a non-persistent viral RNA shedding group. The mean shedding delay times were 47 (31–92 days) and 19 (7–27 days) days for the two groups, respectively. No significant differences were found between the two groups for age, sex, type of hematological malignancy, or COVID-19 evolution. Lymphopenia was an independent predictor of prolonged SARS-CoV-2 RNA detection (*p* = 0.04).

### 3.3. Virological Aspects of COVID in Patients with Hematological Malignancies

Complete sequencing was performed for one hundred and one samples; only the two samples with intermittent positive PCR were not tested. It failed in 21% of cases (n = 22), which was characterized by exceedinga threshold of 32 for the PCR. After a quality sequence analysis, bad sequences were found in four cases: one from the persistent group and three from the non-persistent group. Finally, 75 sequences were analyzed; 46 were obtained from the patients with persistent PCR positivity and 26 from the second group. The mean genome coverage was 98% for all sequences.

Three types of variants were detected: the Alpha (B.1.1.7) variant in one case, the Delta (AY.122.6) in two cases, and the Omicron in thirty-four cases. According to the GENPAT Platform and the Phylogenetic Assignment of Named Global Outbreak Lineages tool (PANGOLIN) nomenclature system, 10 different lineages were detected; BA.1.1 (Omicron) was the most frequent (n = 12; 32.4%) followed by BA.5.2.20 (Omicron) (n = 8; 21.6%). BA.2 (Omicron) was identified in five patients (13.5%) and BA.5.2 (Omicron) in three patients (8.1%). AY.122 (Delta), BA.1.1.1 (Omicron), and BA.1.17.2 (Omicron) were identified in two patients each (5.4%). B.1.1.7 (Alpha), BE.1 (Omicron), and BA.5.2.1 (Omicron) were detected in one patient, respectively (2.7%) (Figure 3). Alpha and Delta variants were detected in the non-persistent viral RNA shedding group; they were associated with an asymptomatic form of infection. The durations of viral RNA positivity were 24, 19, and 17 days, respectively.

In comparisons to the reference sequence (Wuhan-Hu-1 isolate), three types of mutations (nucleotide substitution, nucleotide deletion, and nucleotide insertion) were found distributed throughout the genome. The most frequent were nucleotide substitutions in the spike gene (Figure 4); the principal mutations detected in the studied sequences are reported in Table 3.

Viral evolution was carried out by comparing the first and latest sequences found in the samples for every patient. For 17 patients, sequence analysis showed the inclusion of one to eight amino acid-changes. Genetic stability was shown in six cases across the course of three to twenty days of evolution. The mean intra-host variability was 1.31 × 10^−3^ mutations per site per year globally for all samples and the entire genome; in the persistent group, it was 1.44 × 10^−3^ mutations, while in the non-persistent group, it was 1.3 × 10^−3^ mutations. For the first group, the mean number of mutations was 30 for the spike region and 54 for all genomes, while for the second group, the mean values were 26 and 46, respectively (*p* = 0.650). All observed mutations were described previously in association with SARS-Cov-2 variants circulating during epidemic waves; they were not specific to the studied population. There was no discernible statistically significant difference between the two groups regarding the type of observed mutations (Table 3). Sequence analysis, performed for the two patients with intermittent PCR positivity, shows the same strain persisted intra-host. All mutations identified over all the individual genomes (persistent vs. non-persistent) are reported in Table 1. These mutations served to build the phylogenetic tree reported in Figure 3 as well as the frequency of the mutations through the genomes displayed in Figure 4.

## 4. Discussion

The duration of SARS-CoV-2 persistence following infection is yet unknown, particularly in patients with hematological malignancies. According to our findings, which are the first on this topic in Tunisia as far as we know, 51% of patients with hematological malignancies exhibited ongoing viral shedding. We used a 30-daytime-point after initial positivity in this study, and persistent PCR positivity was taken into consideration regardless of symptomatology. There is currently no consensus on the definition of viral persistence, and the duration of viral shedding is also uncertain. Persistence was defined as the presence of viral RNA for more than 18 days after the onset of disease for symptomatic patients and after the first positive sample for asymptomatic cases in a meta-analysis published by Fontana et al. in 2021 [24]. Some authors have previously reported that a PCR test cannot become negative until 28 days after infection. According to some studies [9,25], a median of 31 days of viral shedding was reported from the beginning of infection for patients with severe forms of infection. In order to investigate the duration of viral excretion in patients with lymphoid malignancies, Lee et al. defined persistent PCR positivity as SARS-CoV-2 RNA detection ≥30 days; persistency was implicated in only 13.9% of their patients [14].

In our study the median duration of PCR positivity was 34 days (range 7–92 days). It has been previously estimated at 59 days (range 26–344) in 13.9% of patients with lymphoid malignancy; most of them (77%) had B-cell non-Hodgkin lymphoma, and at the time of COVID-19 diagnosis, 37% of them were undergoing active work-up or treatment (anti-CD20 monoclonal on 32% of them) [14]. Although the causes of persistent infection are unknown, immunity levels seem to be influenced by several factors. While treatment with anti-CD20 antibodies within a year, cellular therapy (including hematopoietic stem cell transplantation within a year), lymphopenia, and parenteral cytotoxic chemotherapy within six months have all been shown to be independent predictors of prolonged SARS-CoV-2 RNA detection, parenteral cytotoxic chemotherapy within six months was not [26]. The results of our research showed that persistent RNA shedding was substantially correlated with lymphopenia, but not with age, the type of hematological malignancy, treatment category, the presence or absence of comorbidities, neutropenia, or COVID-19 evolution.

Three different SARS-CoV-2 variants were found during the study period; the detection period correlates to the variations’ spread, both locally and internationally. The first and second epidemic waves were linked to the identification of the Alpha variation (B.1.1.7 lineage) in May 2021 and the Delta variant (AY.122) in June–July 2021. The small number of cases reported by these variants can be justified by the preventive measures (blackout and restricted hospital access, only for emergencies) implemented in the country to reduce viral circulation during this period [27]. The fourth and fifth waves were caused by the Omicron variation, which was the most common, and circulated from January 2022 to August 2022 [28,29,30,31,32].

The genetic analysis conducted in this study, as previously reported, confirms the intra-host evolution of the virus and the rate of 1.3 × 10^−3^ mutations reported by other studies in patients with impaired immune systems [12,14]. However, no statistical difference was observed between the two groups, with the same kind of subvariant mutations observed, and at a comparable proportion. It is important to emphasize that observed mutations were linked specifically to the circulating variant [4]. These findings raise the question of whether the persistence of PCR positivity is, really, a consequence of continued viral replication or of the continuous elimination of only viral RNA debris. Then, is it appropriate to continue or to stop immunosuppressive treatment? Is it necessary to isolate patients with persistent viral shedding because of the increased risk of a new variant emerging?

Finally, we did not detect any virological evidence for re-infection; in fact, samples from the two patients who had intermittent PCR positivity were found to have the same genome sequence. Nonetheless, our data validate the intermittent pattern of viral shedding and the requirement for detection of viral RNA in several types of samples in order to avoid the discharge of patients with false negative results [33].

In conclusion, this study highlights the risk of continuous viral shedding in patients suffering from hematological malignancies. However, the stability in viral evolution in patients with persistent PCR positivity and the detection of mostly mutations associated with viral variants suggest that it is probably a continuous elimination of viral RNA debris with a low risk of emergence of a new variant. To confirm this suggestion, it would be advisable to complete this study by the use of viral culture on the samples with prolonged PCR positivity.

## Figures and Tables

**Figure 1 viruses-17-00046-f001:**
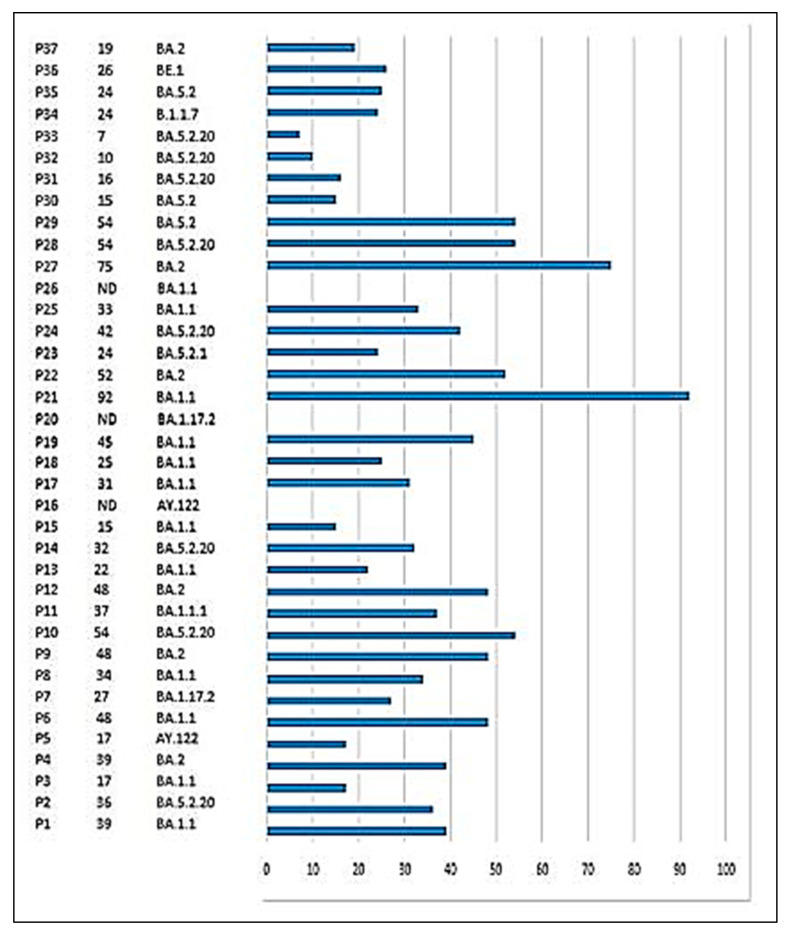
Viral shedding of Sars-cov-2 in hematological malignancy patients (n = 37) and lineage assignment, Tunisia.

**Figure 2 viruses-17-00046-f002:**
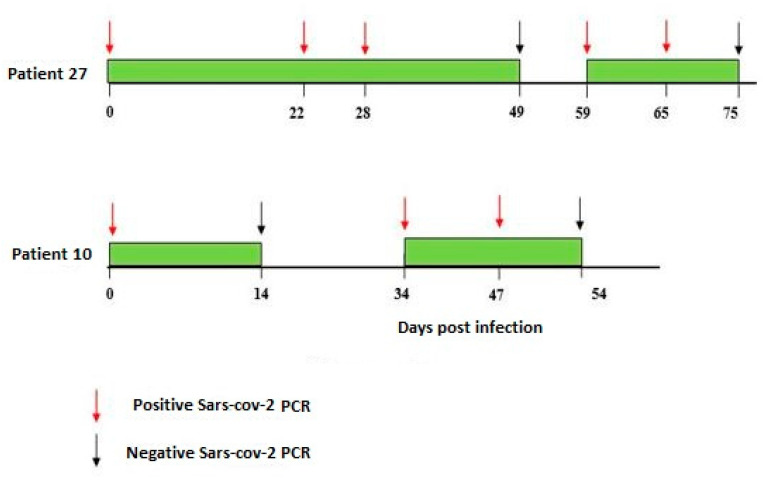
Intermittent PCR positivity in two symptomatic patients.

**Figure 3 viruses-17-00046-f003:**
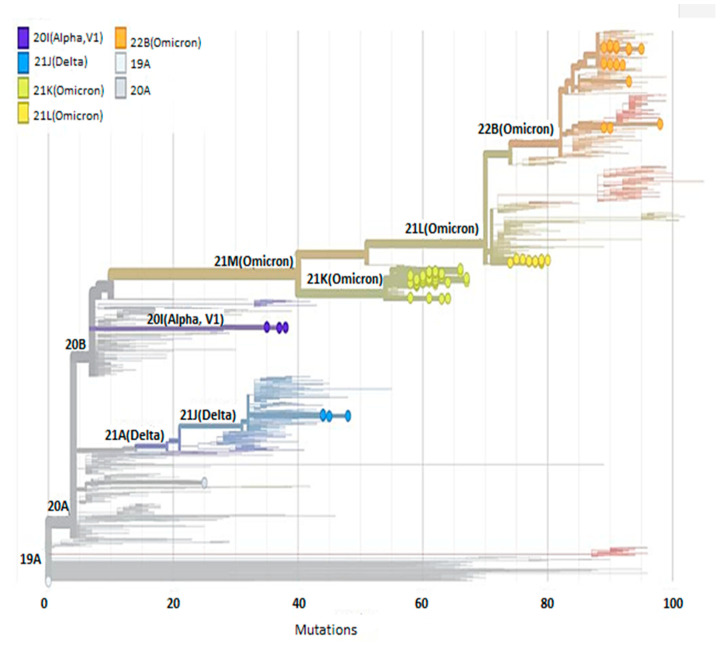
Phylogenetic tree built using the Nextclade online software (https://clades.nextstrain.org/tree, accessed on 2 March 2023) and based on n = 75 severe acute respiratory syndrome coronavirus 2 (SARS-CoV-2) cases detected in Tunisian patients with hematological malignancy [23]. The circles represent the Tunisian sequences in comparison with published sequences from all over the world. Five different clades were detected at the Aziza Othmana Hospital in this type of population: the 20I (Alpha) (n = 1; 2.7%), the 21J (Delta) (n = 2; 5.4%), the 21K (Omicron) (n = 15; 40.5%), the 21L (Omicron) (n = 6; 16.2%), and the 22B (Omicron) (n = 13; 35.1%).The most observed variant was Omicron (n = 34; 91.8%).

**Figure 4 viruses-17-00046-f004:**
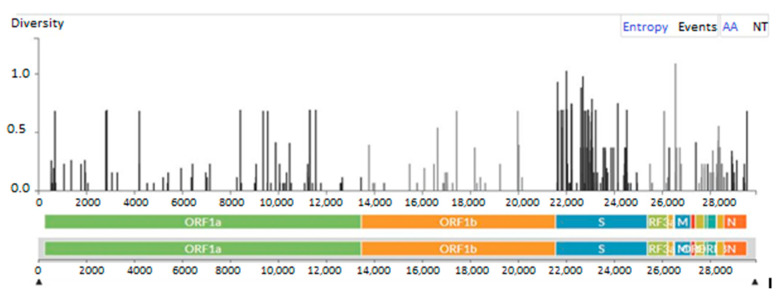
Values of entropy for the changes in AA after analyses of 75 global SARS-COV-2 genomes in Tunisian patients with hematological malignancies.

**Table 1 viruses-17-00046-t001:** Demographic and clinical characteristics of the studied population.

Characteristics	Number of Patients (n, %)
Sex	
Male	: 14 (37.8%)
Female	: 23 (62.2%)
Age	
<18 yearsold	: 10 (27%)
≥18 yearsold	: 27 (73%)
Comorbidities	
Presence	15 (40.5%)
Hypertension	: 8 (21.6%)
Diabetes	: 4 (10.8%)
Others	: 3 (8.1%)
Absence	22 (59.5%)
Type of hematological disease	
Acute Lymphoid Leukemia (LAL)	: 11 (29.7%)
Acute Myeloid Leukemia (LAM)	: 3(8.1%)
Lymphoma	: 15(40.5%)
Myeloma	: 1 (2.7%)
Other hematological diseases	: 7 (18.9%)
SARS-CoV-2 infection	
Asymptomatic form	19 (51.3%)
Symptomatic form	18(48.6%)
Moderate clinical forms	: 32 (86.5%)
Severe clinical forms	: 5 (13.5%)
Biological parameters (Median level)	
C-reactive protein (CRP)	: 27.67 mg/L
PNN	: 2393/mm^3^
Lymphocytes	: 1010/mm^3^

**Table 2 viruses-17-00046-t002:** Demographic and clinical characteristics of persistent viral RNA shedding and for the non-persistent viral RNA shedding group.

Characteristics	Persistent (n = 19, 51.4%) Viral Excretion ≥ 30 Days	Non-Persistent (n = 15, 40.5%) Viral Excretion < 30 Days	*p* Value
Age			
<18 years old	: 6 (31.6%)	4 (26.7%)	0.754
>18 years old	: 13 (68.4%)	11 (73.3%)	
Comorbidities			
Presence:	11 (57.9%)	7 (46.7%)	0.514
Absence	: 8 (42.1%)	8 (53.3%)	
Diagnostic			
Leukemia	: 8 (42.1%)	6 (40%)	0.959
Lymphoma	: 8 (42.1%)	7 (46.6%)	
Other	: 3 (15.8%)	2 (13.3%)	
PNN			
<500/mm^3^	: 5 (26.3%)	6 (40%)	0.397
≥500/mm^3^	: 14 (73.7%)	9 (60%)	
Lymphocytes			
<1000/mm^3^	: 16 (84.2%)	8 (53.4%)	0.04
≥1000/mm^3^	: 3 (16.8%)	7 (47.6%)	
Type of treatment received			
CT + CO	: 10 (52.6%)	8 (53.4%)	0.677
IM + CT + CO	: 2 (10.6%)	3 (20%)	
CT	: 7 (36.8%)	4 (26.6%)	
Type of COVID-19			
Asymptomatic	: 9 (47.4%)	7 (46.6%)	0.967
Symptomatic	: 10 (52.6%)	8 (53.3%)	
COVID-19 Evolution			
Favorable	: 15 (78.95%)	14 (93.3%)	0.962
Death	: 1 (%)	1 (6.7%)	

CT = Chemotherapy; CO = Corticotherapy; IM = Immunotherapy.

**Table 3 viruses-17-00046-t003:** List of the most frequent mutations detected during SARS-CoV-2 infection in the studied population.

Region	Type of Mutation	Frequency (All Patients) (37, 100%)	Frequency (Persistent Group) (19, 51.3%)	Frequency (Non-Persistent Group) (15, 40.5%)
Nsp1	S135R	19, 51.3%	11, 57.9%	8, 53.3%
PLpro	T24I	19, 51.3%	11, 57.9%	8, 53.3%
	G489S	19, 51.3%	11, 57.9%	8, 53.3%
Nsp4	L264F	19, 51.3%	11, 57.9%	8, 53.3%
	T327I	19, 51.3%	11, 57.9%	8, 53.3%
	T492I	35, 94.6%	19, 100%	14, 93.3%
3CLpro	P132H	34, 91.9%	19, 100%	13, 86.7%
Nsp6	Δ106–108	20, 54%	11, 57.9%	9, 60%
RdRp	P323L	37, 100%	19, 100%	15, 100%
Nsp13	T127N	10, 27%	6, 31.6%	4, 26.7%
	R392C	19, 51.3%	11, 57.9%	8, 53.3%
Nsp14	I42V	34, 91.9%	19, 100%	13, 86.7%
Spike	T19I	19, 51.3%	11, 57.9%	8, 53.3%
	L24S	19, 51.3%	11, 57.9%	8, 53.3%
	Δ25–27	19, 51.3%	11, 57.9%	8, 53.3%
	Δ69–70	29, 78.4%	14, 73.7%	13, 86.7%
	G142D	17, 45.9%	9, 47.4%	8, 53.3%
	V213G	19, 51.3%	11, 57.9%	8, 53.3%
	G339D	33, 89.2%	19, 100%	12, 80%
	S371F	32, 86.5%	18, 94.7%	12, 80%
	S375P	32, 86.5%	18, 94.7%	12, 80%
	T376A	19, 51.3%	11, 57.9%	8, 53.3%
	D405N	19, 51.3%	11, 57.9%	8, 53.3%
	R408S	19, 51.3%	11, 57.9%	8, 53.3%
	K417N	32, 86.5%	18, 94.7%	12, 80%
	L452R	16, 43.2%	7, 36.8%	8, 53.3%
	S477N	33, 89.2%	19, 100%	12, 80%
	T478K	35, 94.6%	19, 100%	13, 86.7%
	E484A	32, 86.5%	18, 94.7%	13, 86.7%
	F486V	13, 35.1%	6, 31.6%	7, 46.7%
	Q498R	32, 86.5%	18, 94.7%	12, 80%
	N501Y	33, 89.2%	18, 94.7%	13, 86.7%
	Y505H	32, 86.5%	18, 94.7%	12, 80%
	D614G	37, 100%	19, 100%	15, 100%
	H655Y	34, 91.9%	19, 100%	12, 80%
	N679K	33, 89.2%	18, 94.7%	13, 86.7%
	P681H	37, 100%	19, 100%	15, 100%
	N764K	34, 91.9%	19, 100%	13, 86.7%
	D796Y	34, 91.9%	19, 100%	13, 86.7%
	Q954H	34, 91.9%	19, 100%	13, 86.7%
	N969K	34, 91.9%	19, 100%	13, 86.7%
	T223I	19, 51.3%	11, 57.9%	8, 53.3%
ORF3a	T9I	34, 91.9%	19, 100%	13, 86.7%
E	D3N	27, 73%	13, 68.4%	12, 80%
M	Q19E	34, 91.9%	19, 100%	13, 86.7%
	A63T	34, 91.9%	19, 100%	13, 86.7%
	P13L	34, 91.9%	19, 100%	13, 86.7%
N	Δ31–33	34, 91.9%	19, 100%	13, 86.7%
	R203K	34, 91.9%	19, 100%	14, 93.3%
	G204R	34, 91.9%	19, 100%	14, 93.3%
	S413R	19, 51.3%	11, 57.9%	8, 53.3%

## Data Availability

The raw data supporting the conclusions of this article will be made available by the authors on request.

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
