# Peer review of "Virological Aspects of COVID-19 in Patients with Hematological Malignancies: Duration of Viral Shedding and Genetic Analysis"

_viruses, 2024, doi:10.3390/v17010046_

Round 1
Reviewer 1 Report
Comments and Suggestions for Authors
1. There are so many typos, errors need to be corrected.
2. No patient number in the method section.
3. The implication of this research was not mentioned. What does long term viral shedding mean and how it is important to the hematological malignancies?
4. Overall, the results were not explained properly.
5. The authors stated that "the results of our research showed that persistent RNA shedding was substantially correlated with lymphopenia, but not with age, the type of hematological malignancy, the kind of treatment, the presence or absence of comorbidities, neutropenia, or COVID-19 evolution." (296-98). Please provide with the further proof of this statement. That means the author should provide the experimental evidence that, it is a distinguished observation in patient with COVID-19 and hematological malignancies.
Comments on the Quality of English LanguageQuality of English must be improved.
Reviewer 2 Report
Comments and Suggestions for Authors
Dear authors,
Congratulations on this comprehensive work. I have some simple comments:
1) I suggest one more read-through for editing- for example in the abstract there is a space between words missing
2) I would suggest some added detail in the background more specifically discussing the content of your work- delayed shedding, viral evolution. At present it mainly focuses on the pandemic information which most readers know already. You do include some of this in the discussion. A selection of this information could be moved to background.
3) "the same viral strain had evolved intra-host"- you say this when you are talking about the exact same viral strain being detected twice in same patient- given it is the same strain, I wouldn't say it had evolved- as there is no change. The same strain persisted intra host may be more appropriate to say. Or "we didn't see evidence of evolution of the virus in patients with intermittent viral shedding"
4) I suggest editing the discussion to make your conclusions clearer. It is quite long, but doesn't draw the reader to a clear conclusion in it's current form. What do you think your results mean? Are you confident we don't need to worry about prolonged viral shedding? Try to state your arguments for/against this conclusion more firmly
5) The images are a little blurry in my pdf, please check with the editors regarding whether they need a higher resolution version
6) Please discuss the referencing style with the editors. Vancouver style would be more common in scientific writing generally.
Author Response
Comment N°1: I suggest one more read-through for editing- for example in the abstract there is a space between words missing
Response: We revised the abstract and all the manuscript and added the missing spaces between words as requested
Comment N°2: I would suggest some added detail in the background more specifically discussing the content of your work- delayed shedding, viral evolution. At present it mainly focuses on the pandemic information which most readers know already. You do include some of this in the discussion. A selection of this information could be moved to background.
Response: As requested, we revised the introduction with focusing essentially on viral evolution and duration of SARS-CoV-2 excretion. Also, we revised the discussion; all selected information about viral evolution was removed in “Introduction” section (Lines 51 to 104).
Comment N°3: "the same viral strain had evolved intra-host"- you say this when you are talking about the exact same viral strain being detected twice in same patient- given it is the same strain, I wouldn't say it had evolved- as there is no change. The same strain persisted intra host may be more appropriate to say. Or "we didn't see evidence of evolution of the virus in patients with intermittent viral shedding"
Response: As requested, we changed “the same viral strain had evolved intra-host” in section “Results” lines 271-273 by the following phrase “Sequence analysis, performed for the two patients with intermittent PCR positivity, shows the same strain persisted intra-host.”
Comment N°4: I suggest editing the discussion to make your conclusions clearer. It is quite long, but doesn't draw the reader to a clear conclusion in it's current form. What do you think your results mean? Are you confident we don't need to worry about prolonged viral shedding? Try to state your arguments for/against this conclusion more firmly
Response: As requested, we revised the section “Discussion”: it has shortened to make it clear. We also change the conclusion to make it more precise with giving our opinion more formally based in obtained results (Lines 320 to 342)
Comment N°5: The images are a little blurry in my pdf, please check with the editors regarding whether they need a higher resolution version
Comment N°6: Please discuss the referencing style with the editors. Vancouver style would be more common in scientific writing generally.
Response: For the two latest comments, I ask respectfully the Editor how I can improve the quality of figures and review the referencing style.

Reviewer 3 Report
Comments and Suggestions for Authors
Authors reported duration of SARS-CoV-2 positivity by PCR and evolution of SARS-CoV-2 genome within individual by Whole-genome sequencing in the nasopharyngeal samples from 37 Tunisian patients having hematological malignancies. They concluded (a) the type and the number of mutations were not significantly different between persistent group (viral excretion ≥ 30 days, mean intra-host variability of 1.44 x 10-3) and non-persistent group (viral excretion < 30 days, mean intra-host variability of 1.31 x 10-3); (b) sequences obtained from samples before and after negative PCR confirm an intra-host evolution of the same viral strain; (c) this study confirms the risk of persistent viral shedding in patients with hematological malignancies; and (d) monitoring of the virus's evolution is recommended in this type of population. Although the scale of analyses is remarkable, the usefulness of such frequent sampling and whole genome sequencing in the patients having hematological malignancies remains questionable, especially the conclusion does not appear to provide any incentive to do that. Table 3 shows consistently higher mutation rate in persistent group compared with non-persistent group. It surprises this Reviewer that the difference is not statistically significant. Proper statistical analyses are recommended.
Other concerns:
1. The “up to 30 days” means equal to or less than 30 days. The persistent group, defined as the presence of viral RNA for up to 30 days, would not have viral excretion ≥ 30 days.
2. It is not clear what the “control group” in the Abstract refers to. Also, this study did not include COVID-19 patients without hematological malignancies as control. It is not clear if the duration of shedding is comparable to other Tunisian population.
3. COVID-19 vaccination surely affects the duration. Vaccination status should also be provided.
4. The frequency of sampling and number of samples for whole-genome sequencing in each patient should also be included.
5. The “Section 3.1 Characteristics of patients” belongs to the Materials and Methods, not Results.
6. A discrepancy is found for patients’ ages among Table 1, Figure 1, and Table 2.
7. Extensive correction for grammatical and typographic errors is needed.
Comments on the Quality of English LanguagePerform spelling check and make corrections for English expression.
Author Response
Comment N°1: Authors reported duration of SARS-CoV-2 positivity by PCR and evolution of SARS-CoV-2 genome within individual by Whole-genome sequencing in the nasopharyngeal samples from 37 Tunisian patients having hematological malignancies. They concluded (a) the type and the number of mutations were not significantly different between persistent group (viral excretion ≥ 30 days, mean intra-host variability of 1.44 x 10-3) and non-persistent group (viral excretion < 30 days, mean intra-host variability of 1.31 x 10-3); (b) sequences obtained from samples before and after negative PCR confirm an intra-host evolution of the same viral strain; (c) this study confirms the risk of persistent viral shedding in patients with hematological malignancies; and (d) monitoring of the virus's evolution is recommended in this type of population. Although the scale of analyses is remarkable, the usefulness of such frequent sampling and whole genome sequencing in the patients having hematological malignancies remains questionable, especially the conclusion does not appear to provide any incentive to do that. Table 3 shows consistently higher mutation rate in persistent group compared with non-persistent group. It surprises this Reviewer that the difference is not statistically significant. Proper statistical analyses are recommended.
Response:
- As requested, we revised the conclusion in order to give it concordant with obtained results (Lines 336 to 342 and in the Abstract-Lines 45 to 47).
- The high rate of mutation felt in Table 3 is due to the fact that for we compared our isolates to the Wuhan-HU-1 reference genome (this data is specified in section “bioinformatic analysis” – Lines 150-151). All mutations described for our isolates and reported in Table 3 are specifically linked to circulated variant. We mentioned that in Discussion (Lines 324 – 325); we referred for that to Papanicolaou et al, 2022). To make it more clear, we add in “Results” – Lines 270-271, the following sentence: “All observed mutations were described previously with SARS-Cov-2 variants circulating during epidemic waves”
Other concerns:
Comment N°2: The “up to 30 days” means equal to or less than 30 days. The persistent group, defined as the presence of viral RNA for up to 30 days, would not have viral excretion ≥ 30 days.
Response: Persistent group is characterized by a presence of PCR positivity ≥ 30 days”. To make clear difference between the two groups, we change “up to 30 days” in the manuscript by “more than 30 days” and we change in Table 2, “Viral excretion” by “presence of viral RNA”
Comment N°3: It is not clear what the “control group” in the Abstract refers to. Also, this study did not include COVID-19 patients without hematological malignancies as control. It is not clear if the duration of shedding is comparable to other Tunisian population.
Response:
- To make clearer the abstract, we change in Line 35 “control group” by “non-persistent group”.
- We did not include COVID-19 patients without hematological malignancies as control because the hematological unit in the hospital treats only oncohematology patients. Moreover, prolonged viral excretion was described only for this type of patients (with cancer).
- No data are available about duration of viral shedding in other Tunisian population, that’s why we did not comparison between our population and other groups
Comment N°4: COVID-19 vaccination surely affects the duration. Vaccination status should also be provided.
Response: Unfortunately, vaccination status was not available for our population that’s why we didn’t include it the study. In addition, we did not find previous studies reporting the impact of vaccination on the duration on viral shedding. The question arises whether vaccination can really modify viral shedding.
Comment N°5: The frequency of sampling and number of samples for whole-genome sequencing in each patient should also be included.
Response: As requested, we add frequency of sampling in the “Material and Methods” Lines 115-116 “one sample per week has been planned for each patient”.
Comment N°5: The “Section 3.1 Characteristics of patients” belongs to the Materials and Methods, not Results.
Response: We considered characteristics of patients as results because analysis for this part was performed in the study and we used them to discuss our results “Discussion, Lines 306 – 309)
Comment N°6: A discrepancy is found for patients’ ages among Table 1, Figure 1, and Table 2.
Response: The discrepancy for patient’s age observed in Table 1 and Table 2 is explained by the fact that Table 1 concerned all studied population but Table 2, only patients for whom viral shedding was analyzed. In fact, three patients who died during study were excluded for analysis in Table 2. This data is précised in Results “Characteristics of patients”, Lines 202-203.
Extensive correction for grammatical and typographic errors is needed.
Response: As recommended, Grammar and style were review in the manuscript.

Round 2
Reviewer 3 Report
Comments and Suggestions for Authors
Authors have not yet addressed all previous concerns raised by this Reviewer. The comparison of mutations, for example in Table 3, between the persistent and non-persistent groups should base on individual genome, not individual nucleotide, because each SARS-CoV-2 genome may have no mutation at one site but mutation at other site. Since Figure 3 provides phylogenetic tree, the mutations in each genome should have been identified and should be used for comparison between the persistent and non-persistent groups. The manuscript still recites "control" without specific connection to "non-persistent". Proper statistical analyses are needed.
Comments on the Quality of English LanguageMany grammatical and expression errors remain the major issue for clarity of the manuscript.
Author Response
Tunisia, the 5th december 2024
To the Editor of the journal « Viruses »
Dear Editor,
Please find enclosed the edited and revised manuscript. Please note that all the comments of reviewers were considered and the paper was revised accordingly.
Title: “Virological Aspects of COVID-19 in Patients with
Hematological Malignancies: Duration of Viral Shedding and Genetic Analysis”
Authors: AsmaThemlaoui, Massimo Oncora, Kais Ghedira, Yosra Mhalla, Manel Hamdoun, Maroua Bahri, Lamia Aissaoui, Rayhane Benlakhal, Adriano Di Pasquale, Cesare Camma, Olfa Bahri
Manuscript No: Viruses-3280317
The manuscript has been improved according to the suggestions of reviewers:
"The authors addressed a majority of the concerns raised by the reviewers,
round 1 and round 2. However, there are a few issues that needed to be
addressed before the manuscript is accepted:
- Per report 2 from reviewer #3, the manuscript still recites "control"
without specific connection to "non-persistent". The authors need to correct.
Response: we agree with this point then we reviewed all the manuscript and changed “control group” by “non-persistent group” (Lines 227 and 267).
The tables and figures still appeared blurred. The authors need clear and
concise advice on how to prepare publication quality images. For tables, I
recommend that the authors create those in a word document.
Response: As recommended, Tables were created in a word document and cleared quality of Figures in the manuscript. We also, gave Tables in a separate word document. I hope this version is clearer for Tables and Figures.
3. The authors need to mention in the Methods and/or Results (section 3.1)
that the vaccination status of the patients was not recorded. Also, they need
to comment on whether any of the patients in the study received anti-viral
treatments (monoclonals, anti-viral drugs)."
Response: As recommended, we mentioned in “Results” section that the vaccination status was not recorder. We also add a sentence to give more explanations about patient’s management (Lines 189 to 196). We referred for that, to the National recommendations by experts committee (Reference N°22).
Other comments:
- Ensure all references are relevant to the content of the manuscript.
Response: We verified references; all are relevant to the content of the manuscript
- Highlight any revisions to the manuscript, so editors and reviewers can see any changes made.
Response: Revisions were highlighted as recommended by editors
- Provide a cover letter to respond to the reviewers’ comments and
explain, point by point, the details of the manuscript revisions.
Response: In a cover letter, we responded point by point to given comments
- If the reviewer(s) recommended references, critically analyze them to
ensure that their inclusion would enhance your manuscript. If you believe these references are unnecessary, you should not include them.
Response: No reference was recommended by reviewers for our manuscript
We aslo revised a mistake in the name of one of the author: M. Ancora (instead of M. Oncora)
Olfa Bahri, MD, Professor in Microbiology,
Laboratory of Microbiology and Biochemistry,
Aziza Othmana’s Hospital,
olfa.bahri@fmt.utm.tn

Round 3
Reviewer 3 Report
Comments and Suggestions for Authors
Authors appear to respond to other Reviewer's comments. My comments were "Authors have not yet addressed all previous concerns raised by this Reviewer. The comparison of mutations, for example in Table 3, between the persistent and non-persistent groups should base on individual genome, not individual nucleotide, because each SARS-CoV-2 genome may have no mutation at one site but mutation at other site. Since Figure 3 provides phylogenetic tree, the mutations in each genome should have been identified and should be used for comparison between the persistent and non-persistent groups. The manuscript still recites "control" without specific connection to "non-persistent". Proper statistical analyses are needed.". Satisfactory revision of the manuscript to address this Reviewer's comments is needed.
Comments on the Quality of English LanguageFurther grammatical and expression correction is needed.
Author Response
Comment:
Authors appear to respond to other Reviewer's comments. My comments were "Authors have not yet addressed all previous concerns raised by this Reviewer. The comparison of mutations, for example in Table 3, between the persistent and non-persistent groups should base on individual genome, not individual nucleotide, because each SARS-CoV-2 genome may have no mutation at one site but mutation at other site. Since Figure 3 provides phylogenetic tree, the mutations in each genome should have been identified and should be used for comparison between the persistent and non-persistent groups. The manuscript still recites "control" without specific connection to "non-persistent". Proper statistical analyses are needed.". Satisfactory revision of the manuscript to address this Reviewer's comments is needed.
Response:
As recommended by Reviewer, we add to the manuscript a supplementary Table 1 which reports all mutations identified over all the individual genomes (persistent vs non-persistent). We mentioned also this before Table 3, in Lines 274 to 279 as follows: “All mutations identified over all the individual genomes (persistent vs non-persistent) are reported in supplementary table 1. These mutations served to build the phylogenetic tree reported in Figure 3 as well as the frequency of the mutations through the genomes displayed in Figure 4”.
However, Table 3 lists the mutations identified by the sequencing analysis performed on our positive samples. As we mentioned in Lines 268 to 273, none of these mutations are specific to our studied population (patients with hematological malignancies). All these mutations were previously described for circulating variants during the studied period (epidemic waves caused by Omicron or Delta variants). Also, there was a statistically significant difference between the two groups.